# The miRNA *bantam* regulates growth and tumorigenesis by repressing the cell cycle regulator *tribbles*

Stephan U Gerlach, Moritz Sander, Shilin Song, Héctor Herranz

One of the fundamental issues in biology is understanding how organ size is controlled. Tissue growth has to be carefully regulated to generate well-functioning organs, and defects in growth control can result in tumor formation. The Hippo signaling pathway is a universal growth regulator and has been implicated in cancer. In *Drosophila*, the Hippo pathway acts through the miRNA *bantam* to regulate cell proliferation and apoptosis. Even though the *bantam* targets regulating apoptosis have been determined, the target genes controlling proliferation have not been identified thus far. In this study, we identify the gene *tribbles* as a direct *bantam* target gene. Tribbles limits cell proliferation by suppressing G2/M transition. We show that *tribbles* regulation by *bantam* is central in controlling tissue growth and tumorigenesis. We expand our study to other cell cycle regulators and show that deregulated G2/M transition can collaborate with oncogene activation driving tumor formation.

## Introduction

Tissue growth has to be precisely regulated to generate organs with correct size, shape, and function. However, how growth is controlled remains a fundamental question in biology. In most animals, growth is the result of the increase in the number of cells, which is determined by the rate of cell division and cell death (Conlon & Raff, 1999). Deregulation in those processes can result in the formation of tumors (Hanahan & Weinberg, 2011). The wing imaginal disc of *Drosophila* has proven to be a useful system to study tissue growth control and to model different aspects related to tumor formation and metastasis (Gonzalez, 2013; Hariharan, 2015; Herranz et al, 2016).

miRNAs are small noncoding RNAs that have emerged as central regulators in the expression and function of animal genomes. miRNAs regulate multiple processes in development, including tissue growth, stem cell development, hormone action, and the organization and function of the central nervous system (Carthew et al, 2017). In addition, miRNAs control the activity of oncogenes and tumor suppressors and, thus, play crucial roles in cancer initiation and disease progression (Esquela-Kerscher & Slack, 2006). Understanding the roles of miRNAs in development and cancer requires the identification of the target genes responsible for their functions.

The Hippo tumor suppressor pathway is a universal growth regulator and has been implicated in cancer (Pan, 2010). In *Drosophila*, the Hippo pathway consists of a cascade of kinases, which function sequentially to restrict the nuclear localization of the growth-promoting transcriptional coactivator, Yorkie (Yki) (Huang et al, 2005). Yki regulates the expression of the miRNA *bantam* and *bantam* mediates its growth-promoting role (Nolo et al, 2006; Thompson & Cohen, 2006). *bantam* was the first miRNA discovered in *Drosophila*. It was identified as an element that, when overexpressed in the wing, promotes tissue growth (Hipfner et al, 2002; Brennecke et al, 2003). *bantam* stimulates growth by inducing proliferation and inhibiting apoptosis (Brennecke et al, 2003). *bantam* can also cooperate with the oncogene *epithelial growth factor receptor* (*EGFR*) in the formation of neoplastic tumors (Herranz et al, 2012b). Years of extensive research have led to the identification of multiple *bantam* targets, including *suppressor of cytokine signaling at 36E (Socs36E)*, *head involution defective (hid)*, *capicua (cic)*, *meiotic P26* (*mei-P26*), and *enabled (ena)* (Brennecke et al, 2003; Herranz et al, 2010, 2012a, 2012b; Becam et al, 2011). However, the current knowledge of the *bantam* target genes remains incomplete because individual depletion of any of them fails to mimic the effect of *bantam* driving tissue growth.

Here, we identify the gene *tribbles* (*trbl*) as a direct *bantam* target gene that mediates its growth regulatory role. We find that *trbl* downregulation in the wing disc accelerates G2/M progression in the cell cycle. We show that simultaneous depletion of the *bantam* targets *trbl* and the proapoptotic gene *hid* reproduces the effect of *bantam*, promoting growth. The results presented here determine the minimal combination of *bantam* target genes sufficient to mimic the effect of *bantam* in tissue growth. We provide evidence that Trbl not only mediates the *bantam* growth-promoting role but also limits the magnitude of tissue overgrowth induced by the proto-oncogene Yki. Finally, we find that depletion of *trbl* can drive tumorigenesis in different contexts. Taken together, these results demonstrate that, in some instances, Trbl can limit tumorigenesis in the imaginal epithelium. In summary, our study demonstrates that *trbl* is a crucial target mediating the growth-promoting role and oncogenic function of *bantam*.

Department of Cellular and Molecular Medicine, University of Copenhagen, Copenhagen, Denmark

Correspondence: hherranz@sund.ku.dk

## Results

### bantam represses trbl

miRNAs are negative regulators of gene activity. They bind to complementary sites on target mRNAs to induce mRNA cleavage or to repress mRNA translation. Understanding the oncogenic and growth regulatory roles of *bantam* requires the identification of its target genes. Bioinformatic analysis predicted two *bantam*-binding sites in the 3'UTR of the gene *trbl* (Fig 1A). *trbl* encodes a kinase-like protein that controls cell proliferation by regulating the stability of the cell cycle regulator, cdc25-stg (Mata et al, 2000).

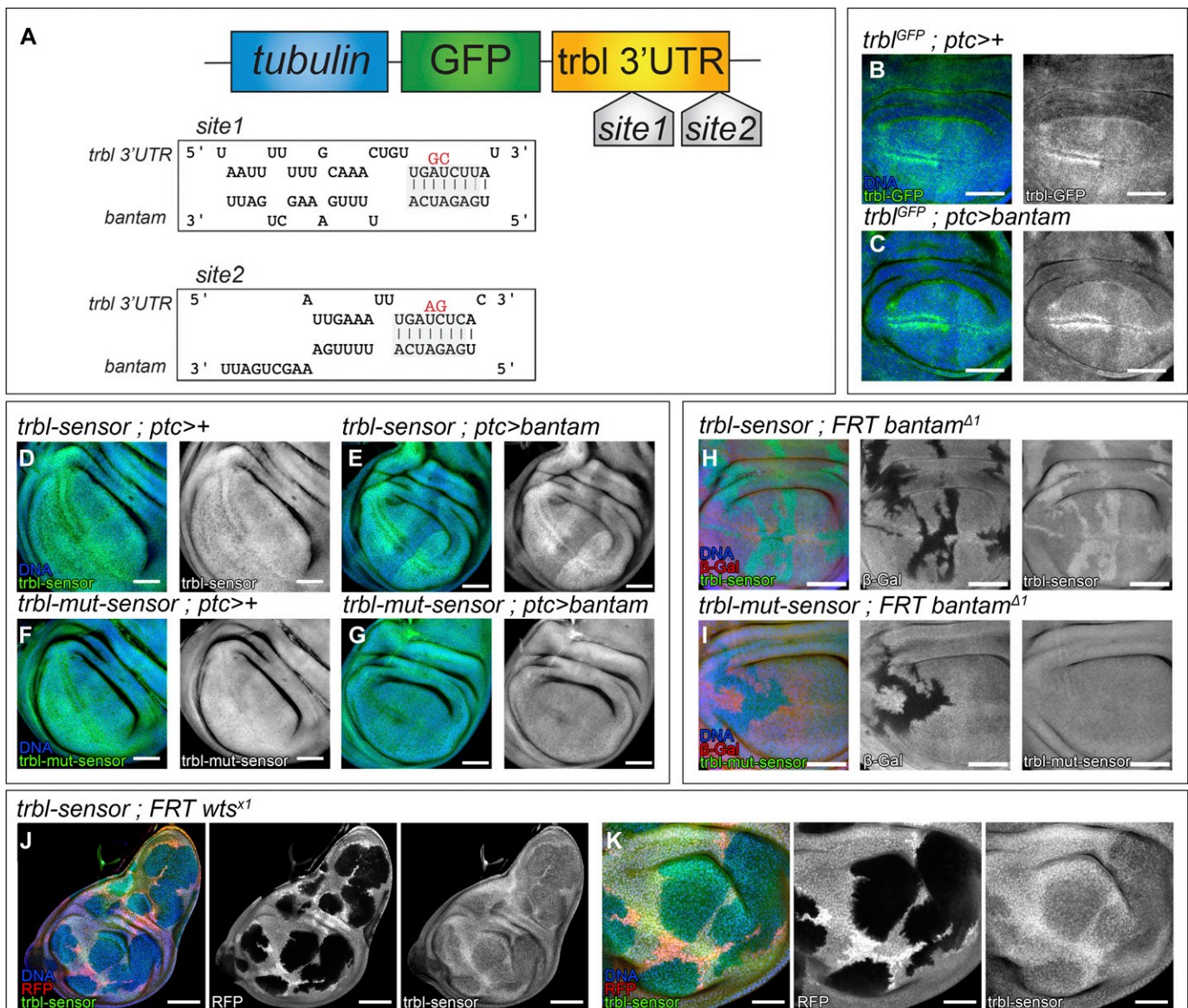

**Figure 1. *bantam* represses *trbl* directly.**
**(A)** Illustration of *trbl-sensor* showing predicted *bantam*-binding sites. The transgene consists of a tubulin promoter expressing the mRNA for GFP fused to the *trbl* 3'UTR. The mRNA is sensitive to regulation through regulatory elements in the *trbl* 3'UTR. The *trbl* 3'UTR includes two predicted binding sites for *bantam* (indicated as site 1 and 2). The predicted binding between the nucleotides of the *trbl* 3'UTR and *bantam* is shown, and the predicted seed sequences are highlighted (connecting lines between the nucleotides and grey background). The *trbl-mut-sensor* has two nucleotides exchanged in each of the predicted seed sequences (highlighted in red). **(B, C)** Confocal micrographs of third-instar wing imaginal discs showing *trbl-GFP* expression (green and grey) in the genotypes indicated. DAPI was used to label the nuclei (blue). Larvae were kept constantly at 25°C. Scale bar: 50 µm. **(D–G)** Confocal micrographs of third-instar wing imaginal discs showing the expression of *trbl-sensor* (D, E) and *trbl-mut-sensor* (F, G) in the genotypes indicated. The sensors are shown in green and grey. DAPI was used to label the nuclei (blue). Larvae were kept constantly at 25°C. Scale bar: 50 µm. **(H, I)** *bantam*[Δ1]-mutant clones marked by the absence of β-Gal (red and grey), showing the expression of the *trbl-sensor* in (H) (green and grey), and showing the expression of the *trbl-mut-sensor* in (I) (green and grey). DAPI was used to label the nuclei (blue). To facilitate the analysis, the clones were induced in a *Minute*/+ background to allow the recovery of bigger clones. Scale bar: 50 µm. **(J, K)** *wts*[x1]-mutant clones marked by the absence of RFP (red and grey), showing the expression of the *trbl-sensor* (green and grey). DAPI was used to label the nuclei (blue). (K) Shows a magnification of the wing disc shown in (J). Scale bar: 100 µm in (J) and 50 µm in (K).

To validate a miRNA-target prediction, it is necessary to correlate the expression of miRNAs with those of their potential targets. miRNAs are posttranscriptional regulators and, to be functional, need to be co-expressed with their targets. To detect Trbl protein, we used a GFP-insertion in the *trbl* locus, resulting in a Trbl-GFP fusion protein (Mi(Mic)trbl-MI01025) (Venken et al, 2011). We monitored *bantam* expression by using a P element insertion (*bantam-lacZ*) that reports expression from the *bantam* locus (Herranz et al, 2012a). Interestingly, Trbl and *bantam* were both observed in the wing disc, indicating that the bioinformatical prediction might reveal a functional interaction (Fig S1).

Next, we tested whether *bantam* up-regulation affected Trbl levels. Trbl showed an accumulation in two rows of cells abutting the dorsal–ventral boundary in the anterior compartment of the wing primordia (Fig 1B). Interestingly, Trbl limits G2/M progression (Mata et al, 2000) and the cells showing the highest Trbl levels correspond to the precursors of the sensory organs in the adult wing margin, which are arrested in G2 (Johnston & Edgar, 1998). We made use of the binary Gal4:UAS system to manipulate *bantam* levels in the wing disc (Brand & Perrimon, 1993). *patched-Gal4* (*ptc-Gal4*) is expressed in a stripe of cells adjacent to the anterior–posterior and perpendicular to the dorsal–ventral border (Fig S1). Overexpression of *UAS-bantam* in the *ptc* domain reduced Trbl levels, as compared with the surrounding normal cells (Fig 1C). We obtained comparable results using the *EP3622* element (Fig S1), an EP line inserted in the *bantam* locus that, when combined with a Gal4 driver, directs the expression of *bantam* (Hipfner et al, 2002). Consistent with those observations, Trbl-GFP levels were increased in *bantam* mutants (Fig S1). These results, coupled with the identification of *bantam*-binding sites in the *trbl* 3′UTR and the observations that *trbl* and *bantam* are co-expressed, suggest that *trbl* is a potential *bantam* target gene.

### *trbl* is a direct *bantam* target

To assess whether *bantam* regulates *trbl* directly, we cloned the *trbl* 3′UTR downstream of a tubulin promoter-EGFP reporter plasmid (*trbl-sensor*, Fig 1A). The *trbl-sensor* was expressed ubiquitously in the wing imaginal disc (Fig 1D). Consistent with our previous observations, *bantam* overexpression in the *ptc* domain led to a robust reduction of the *trbl-sensor* (Fig 1E). To test whether *trbl* regulation by *bantam* occurred through the predicted binding sites, we generated a mutant version of the *trbl-sensor* (*trbl-mut-sensor*), carrying modifications in the both binding sites (illustrated in Fig 1A). Remarkably, the *trbl-mut-sensor* was not sensitive to *bantam* overexpression (Fig 1F and G). Overexpression of *bantam* using *EP-bantam* (*EP3622*) yielded similar results, yet the regulation over the *trbl-sensor* was lower in magnitude than the one observed when expressing *UAS-bantam* (Fig S1).

Next, we studied whether the *trbl* 3′UTR was sensitive to endogenous *bantam* levels. To perform that analysis, we generated *bantam* null mutant clones (*FRT bantam-Δ1*) in the wing disc. *bantam* mutant clones grow poorly and are eliminated from the wing epithelium presumably by cell competition (Thompson & Cohen, 2006). To bypass this issue, we provided the *bantam* mutant clones with a competitive advantage over neighboring cells by inducing the clones in a *Minute/+* background (Morata & Ripoll,

1975). The *bantam* mutant clones were marked by the absence of *lacZ* expression. *bantam* mutant cells up-regulated the *trbl-sensor* (Fig 1H). In contrast to that, the expression of the *trbl-mut-sensor* was not affected in *bantam* clones (Fig 1I), providing additional evidence that *trbl* regulation by *bantam* is direct. The *trbl-sensor* was not affected in *+/+ control* clones induced in a *Minute/+* background (Fig S2). In summary, these observations indicate that *trbl* is a direct *bantam* target gene and show that the *trbl* 3′UTR is sensitive, not only to *bantam* overexpression but also to endogenous *bantam*.

The Hippo pathway regulates the expression of *bantam*, which is central in mediating the Yki growth regulatory role (Nolo et al, 2006; Thompson & Cohen, 2006). Hippo activation results in Warts (Wts) phosphorylation, which acts with its cofactor Mats to phosphorylate and inactivate Yki. *wts*-mutant tissue up-regulates Yki and overgrows (reviewed in (Pan, 2010)). As observed in wing discs up-regulating *bantam*, *wts* mutant cells down-regulated the *trbl-sensor* (Fig 1J and K). The *trbl* 3′UTR is, thus, sensitive to Hippo pathway activity.

### *trbl* antagonizes G2/M transition in the wing disc

The conserved gene *trbl* was first identified in *Drosophila* as a gene coordinating embryonic cell division and morphogenesis (Mata et al, 2000). In the wing disc, cells overexpressing *trbl* show longer G2/M phases and accumulate Cyclin E (Mata et al, 2000; Reis & Edgar, 2004). However, the consequences of down-regulating *trbl* in the wing disc and its effects in growth control and tumorigenesis are not yet determined.

The fluorescence ubiquitin cell cycle indicator method (FUCCI) is a two-color sensor that provides a readout of the cell cycle phase of each cell in a population (Sakaue-Sawano et al, 2008). *Drosophila* FUCCI (Fly-FUCCI) marks cells in the G1-phase in green, cells in the S-phase (DNA synthesis) in red, and cells in the G2-phase in yellow (Fig 2A) (Zielke et al, 2014). We combined the use of Fly-FUCCI with flow cytometry–based cell cycle analysis to study the cell cycle dynamics in discs modulating *trbl*. Expression of *UAS-Fly-FUCCI* in normal discs showed cells labeled in red, green, and yellow, indicating the presence of proliferating cells (Fig 2B). *trbl* overexpression led to an increase in cells labeled in yellow, representing an accumulation of cells in G2 (Fig 2C). We used anti-phospho-Histone H3 (PH3) as a specific marker of cells in mitosis to study whether *trbl* overexpression affected the rate of cell proliferation of the wing disc. We detected a reduction in the number of PH3-positive cells in discs expressing *UAS-trbl* when compared with control discs (Fig 2E–G). These observations are consistent with previous reports (Mata et al, 2000; Reis & Edgar, 2004) and show that *trbl* overexpression limits cell proliferation by dampening G2/M progression.

Next, we evaluated the role of endogenous *trbl* in the wing disc. Depletion of *trbl* by expression of a *UAS-trbl-RNAi* transgene caused a reduction in the number of cells marked in yellow, which indicates that those cells show faster G2/M transition (Fig 2D). The efficiency of the *UAS-trbl-RNAi* transgene is shown in Fig S3. In sum, these results confirm previous observation and reveal that endogenous *trbl* limits G2/M progression in the wing disc.

We used the adult wing to analyze how *trbl* affected cell number and tissue size. In the adult wing, each hair-like structure corresponds

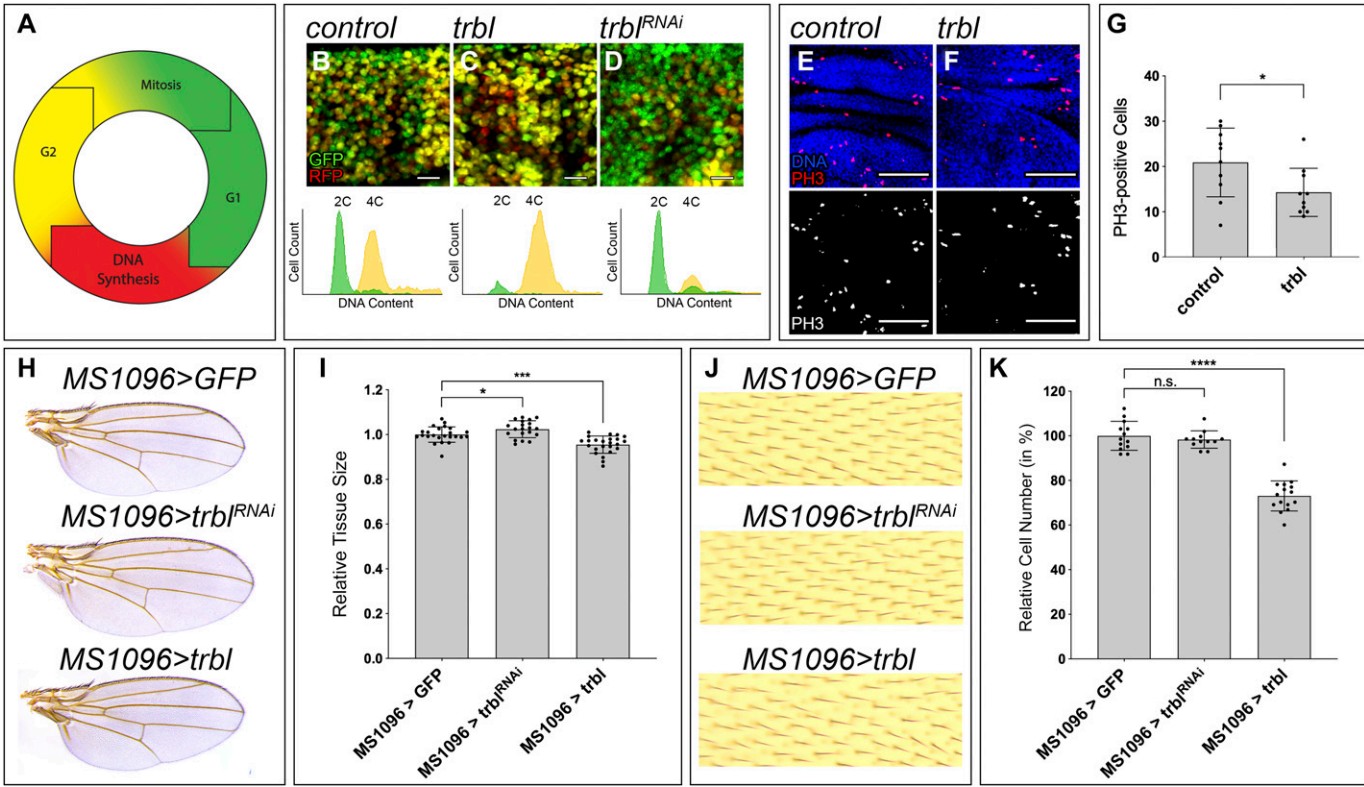

**Figure 2. *trbl* phenotype in the wing disc.**
**(A)** Illustration of fly-FUCCI system. Cells are labeled in green in late mitosis and the G1 phase, in red during the S-phase (DNA synthesis), and in yellow in the G2 phase and early mitosis. **(B–D)** Third-instar wing imaginal discs expressing the fly-FUCCI system together with *UAS-trbl* (C) and *UAS-trbl-RNAi* (D). The indicated genes were expressed under the control of the *ap-Gal4* driver. *ap-Gal4:UAS-FUCCI* was used as control (B). Scale bar: 10 *μm*. Bottom panels show flow cytometry cell cycle analysis by quantification of DNA content in the genetic backgrounds shown in (B–D). Flow cytometry and fly-FUCCI were combined and the corresponding profiles are shown in the bottom of each panel. In these profiles, 2C and 4C DNA contents are indicated, and the cells in G1 (green) and G2 (yellow) are highlighted. **(E, F)** Confocal micrographs of third-instar wing imaginal discs showing mitotic cells labeled by anti-PH3 (red and grey) in the following genotypes: *ap-Gal4:UAS-GFP* (E) and *ap-Gal4:UAS-GFP; UAS-trbl* (F). DAPI was used to label the nuclei (blue). Scale bars: 50 *μm*. **(G)** Quantification of PH3-positive cells in third-instar wing imaginal discs. Examples of quantified area in (E, F). Statistical significance was determined by unpaired *t* test (n = 10; *P < 0.05). **(H)** Cuticle preparations of adult wings of the genotypes indicated. **(I)** Quantification of adult wing size. Wing area is normalized to the mean of the control. Statistical significance was determined by unpaired *t* test (n = 24 [MS1096>GFP], n = 20 [MS1096>trbl-RNAi], n = 24 [MS1096>trbl]; *P < 0.05; ***P < 0.001). **(J)** High-magnification image showing cuticle preparations of adult wings of the genotypes indicated. Note the reduction in cell density reflecting the presence of bigger cells in the discs overexpressing *trbl* (*MS1096>trbl*). **(K)** Quantification of cell number in adult wings. Examples of quantified area in (J). Cell number is normalized to the mean of the control. Statistical significance was determined by unpaired *t* test (n = 13 [MS1096>GFP], n = 12 [MS1096>trbl-RNAi], n = 15 [MS1096>trbl]; ns, not significant; ****P < 0.0001).

to a trichome originating from a single epithelial cell. Hence, trichome density can be used as a proxy of cell size and number. *trbl* depletion did not affect cell or wing size. In contrast, *trbl* overexpression caused a mild reduction in wing size. These wings showed an increase in cell size revealed by reduced trichome density, indicating that *trbl*-overexpressing wings had fewer cells (Figs 2H–K and S4). The increased cell size observed upon *trbl* up-regulation is characteristic of cells with delayed cell cycle pro-gression (Neufeld et al, 1998) and is consistent with the PH3 staining and the FUCCI-FACS analysis.

### *trbl* is involved in the growth regulatory role of *bantam*

The fact that Trbl regulates the cell cycle in the wing disc indicates that this gene is a promising candidate mediating the role of *bantam*, inducing cell proliferation. If *trbl* repression by *bantam*

contributes to the growth-promoting role of this miRNA, we rea-soned that restoring *trbl* expression in discs expressing *bantam* should reduce the magnitude of tissue overgrowth. Although *trbl* overexpression did not overtly affect the size of the wing disc, *bantam*-induced overgrowth was reduced by expression of *UAS-trbl* in place of the *UAS-lacZ* control (Fig 3A–D and G). Taken to-gether with the finding that *bantam* regulates *trbl*, these results suggest that *bantam* acts, at least partially, through *trbl* to stimulate tissue growth.

Given that *bantam* is an essential Yki target mediating its growth regulatory role (Nolo et al, 2006; Thompson & Cohen, 2006), we reasoned that Trbl should also restrict the growth-promoting effect of Yki. Indeed, co-expression of *trbl* with *yki* was sufficient to limit the magnitude of tissue overgrowth induced by *yki* on its own (Fig 3E–G). It is worth noting that *trbl* expression in an otherwise normal background did not have a strong impact on the size of the wing

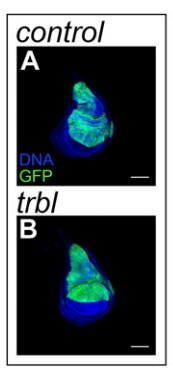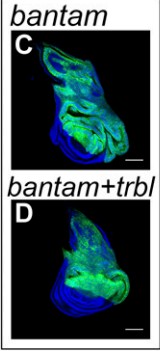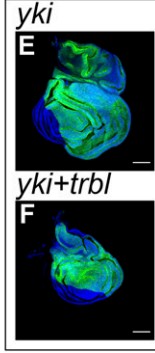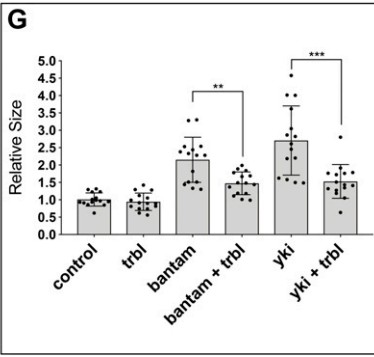

**Figure 3.** *trbl* dampens *bantam*- and Yki-induced tissue growth.
**(A–F)** Confocal micrographs of third-instar wing imaginal discs. The indicated genes were expressed under the control of the *ap-Gal4* driver (dorsal compartment). The discs also expressed *UAS-GFP* to label the region where the corresponding transgenes were induced (green). DAPI was used to label the nuclei (blue). *ap-Gal4:UAS-GFP* was used as control (A). Scale bar: 100 µm. **(G)** Quantification of GFP-positive area in third-instar wing imaginal discs. GFP-positive area is normalized to the mean of the control. Statistical significance was determined by unpaired *t* test (n = 15; **P < 0.01; ***P < 0.001).

imaginal tissue (Fig 3B and G), indicating that the growth-repressing function of *trbl* is specific to the Yki and *bantam* axis, and not a general consequence of *trbl* overexpression on normal tissue growth.

### *bantam* promotes tissue growth through *trbl* and *hid*

The results shown previously suggest that *trbl* is a central *bantam* target in growth control. Although *trbl* knock down in the wing disc accelerated G2/M progression, it did not result in obvious tissue overgrowth (Fig 4A, B, and E). This suggests that *trbl* is not the only gene through which *bantam* induces tissue overgrowth. Changes in cell proliferation in the imaginal discs are typically associated with an increase in cell death (Karim & Rubin, 1998; Neufeld et al, 1998). In good agreement, the use of an antibody recognizing the activated form of caspase 3 revealed the presence of apoptotic cells upon *trbl* depletion (Fig 4F–H). *bantam* inhibits cell death by repressing the proapoptotic gene *hid* (Brennecke et al, 2003). As in the case of *trbl*, *hid* down-regulation did not affect the size of the wing disc (Fig 4C and E).

We have shown that independent depletion of the *bantam* targets *trbl* or *hid* did not mimic the *bantam* gain of function phenotype. A cooperative interaction between both genes might mediate the growth-promoting role of *bantam*. We hypothesized that, to promote overall tissue growth, *bantam* could use a dual mechanism: on the one hand, it would accelerate G2/M transition by repressing *trbl* and, on the other hand, it would inhibit apoptosis by repressing the proapoptotic gene *hid*. If this is true, replacing *bantam* expression by co-depletion of both targets should lead to a similar outcome. To test this hypothesis, we expressed simultaneously *UAS-RNAi* transgenes directed against both *bantam* targets: *UAS-trbl-RNAi* and *UAS-hid-RNAi*. Remarkably, concurrent depletion of *trbl* and *hid* led to the formation of overgrown imaginal discs (Fig 4D and E), resembling *bantam* overexpression. *hid* depletion did not completely rescue the induction of apoptosis observed in wing discs expressing *UAS-trbl-RNAi* (Fig S5), suggesting that other proapoptotic genes, in addition to *hid*, might contribute to the apoptotic response to *trbl* knock down.

Next, we studied the consequences of down-regulating *trbl* and *hid* in discs with reduced *bantam*. miRNA sponges controlled by *UAS* sequences allow spatially controlled down-regulation of

miRNAs under Gal4 control (Loya et al, 2009). The expression of a *UAS-bantam-sponge* transgene causes a reduction in tissue size (Herranz et al, 2012a). Notably, discs expressing simultaneously *UAS-bantam-sponge*, *UAS-trbl-RNAi*, and *UAS-hid-RNAi* were increased in size (Fig 4I–K). This suggests that co-depletion of *trbl* and *hid* is sufficient to drive tissue overgrowth, even when *bantam* levels are reduced.

We have shown that *trbl* and *hid* are important targets contributing to the growth-promoting role of *bantam*. Tumor formation appears to require both an increase in cell division and a suppression of apoptosis (Green & Evan, 2002). The apoptotic response observed in wing discs depleting *trbl* might serve as a homeostatic process to regulate organ size and prevent the formation of tissue hyperplasia. To assess that, we expressed the baculovirus protein p35 (Hay et al, 1994) or a miRNA that inhibits the proapoptotic genes *reaper*, *hid*, and *grim* (*miR-RHG*) (Siegrist et al, 2010), as alternative means to repress apoptosis. Remarkably, suppression of apoptosis in cells depleting *trbl* also led to tissue overgrowth (Fig S5). The use of an independent *UAS-trbl-RNAi* line produced comparable results (Fig S5). Similar results were obtained when we overexpressed *cdc25-stg* as an alternative method of forcing G2/M transition (Fig S6). In sum, these results suggest that apoptosis in response to *trbl* depletion can be used as a homeostatic mechanism to offset tissue overgrowth.

Next, we studied cell proliferation in discs co-expressing *p35* and *trbl-RNAi*. We used proliferation markers such as 5-ethynyl-20-deoxyuridine (EdU), which marks cells in the S-phase and PH3 to label cells in mitosis. This revealed an increase in the cell division rate in cells depleting *trbl* and expressing *p35*, as compared with the normal cells in the ventral compartment (GFP-negative) that served as internal control (Fig S5). In conclusion, these results reflect that simultaneous induction of G2/M and suppression of apoptosis causes tissue hyperproliferation and hyperplasia.

### Trbl suppresses the oncogenic function of *bantam*

EGFR is a well-recognized cancer driver. Overexpression of wild-type *EGFR* in *Drosophila* causes the activation of the MAPK pathway and tissue hyperplasia (Herranz et al, 2012a). Cooperating factors are required for the development of neoplastic tumors. *bantam* has been identified as one of such factors, whereby *bantam* inhibits the

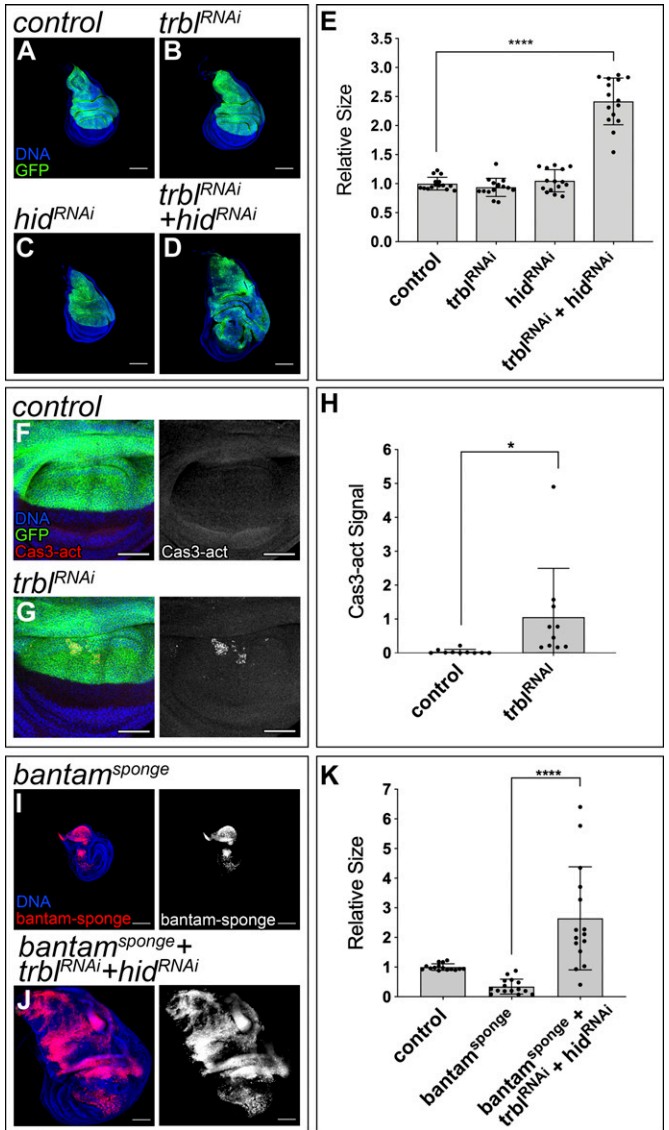

**Figure 4.  *trbl* and *hid* are key *bantam* targets in growth control.**
**(A–D)** Confocal micrographs of third-instar wing imaginal discs. The indicated genes were expressed under the control of the *ap-Gal4* driver (dorsal compartment). The discs also expressed *UAS-GFP* to label the region where the corresponding transgenes were induced (green). DAPI was used to label the nuclei (blue). *ap-Gal4:UAS-GFP* was used as control (A). Scale bar: 100 *µm*. **(E)** Quantification of GFP-positive area in third-instar wing imaginal discs. GFP-positive area is normalized to the mean of the control. Statistical significance was determined by unpaired *t* test (n = 15; ****$P < 0.0001$). **(F, G)** Third-instar wing imaginal discs showing staining against Cas3-activated signals (red and grey). DAPI was used to label the nuclei (blue). The indicated genes were expressed under the control of the *ap-Gal4* driver (dorsal compartment). The discs also expressed *UAS-GFP* to label the region where the corresponding transgenes were induced (green). *ap-Gal4:UAS-GFP* was used as control (F). Larvae were kept constantly at 25°C. Scale bar: 50 *µm*. **(H)** Quantification of Cas3-activated signal in the wing pouch of third-instar wing imaginal discs. The signal is represented by the mean grey value. Statistical significance was determined by *t* test (n = 10; *$P < 0.05$). Larvae were kept constantly at 25°C. **(I, J)** Third-instar wing imaginal discs expressing *UAS-bantam-sponge-RFP* (I) and *UAS-bantam-sponge-RFP*, *UAS-trbl-RNAi*, and *UAS-hid-RNAi* (J), under the control of the *ap-Gal4* driver (dorsal compartment). Cells expressing *bantam-sponge-RFP* are labeled in red. DAPI was used to label the nuclei (blue). Larvae were kept constantly at 25°C. Scale bar: 100 *µm*. **(K)** Quantification of the dorsal compartment size in third-instar wing imaginal discs. The control expressed GFP under the control of *ap-*

suppressor of cytokine signaling *Socs36E*, which limits JAK/STAT signaling and tumor development (Herranz et al, 2012b).

We studied whether *trbl* also contributed to the *bantam* oncogenic function. Indeed, *trbl* expression in *EGFR + bantam* tumors impaired tumor growth leading to the formation of discs comparable in size with the one observed in discs expressing *EGFR* on its own (Fig 5A–C and G). This suggests that *trbl* not only limits the *bantam* growth-promoting role, but also reduces its oncogenic effect. Next, we analyzed whether *trbl* down-regulation is sufficient to cooperate with the oncogene *EGFR* in tumorigenesis. Interestingly, although *trbl* depletion did not have a big impact in normal growth, *trbl* knock down in a context of *EGFR* overexpression fueled tumor growth (Fig 5D and G). These observations indicate that *trbl* behaves as a tumor suppressor in a context of *EGFR* expression.

Next, we studied the consequences of down-regulating other G2/M repressors in a context of EGFR activation. Activation of the Cdk1–Cyclin B complex induces cells to enter mitosis from G2. Myt1 is the main Cdk1 inhibitory kinase in the wing imaginal disc (Jin et al, 2008). Consistent with our observations in cells knocking down *trbl*, the expression of *UAS-Myt1-RNAi* accelerated G2/M transition and did not have an overt effect in disc size (Fig S7). Importantly, *Myt1* depletion cooperated with EGFR in tumor formation (Fig 5E and G). Similar results were obtained when we drove G2/M transition by expressing *cdc25-stg* in discs up-regulating *EGFR* (Fig 5F and G). Together, this reveals that although deregulation in G2/M transition does not have a major impact in normal tissue growth, it can be sufficient to trigger tumorigenesis in combination with the onco-gene EGFR.

Trbl, in addition to regulate cell cycle progression, limits growth by antagonizing the insulin receptor/phosphatidylinositol 3-kinase (PI3K)/AKT protein kinase pathway (Das et al, 2014; Hong et al, 2016). This signaling cascade is hyperactivated in a wide range of human cancers (Vivanco & Sawyers, 2002). Even though inducing G2/M appears to be sufficient to cooperate with EGFR in tumorigenesis (Fig 5D–G), we cannot rule out that PI3K/AKT regulation by *trbl* might also contribute to its tumor suppressor function.

## Trbl tumor suppressive role in other tumor contexts

Cytokinesis failure leads to the formation of tetraploid cells. This generates an unstable situation that can lead to aneuploidy and tumor formation after cell division (Ganem et al, 2007). We have recently developed a genetic model in the wing imaginal disc of *Drosophila* to study the oncogenic potential of cytokinesis failure. The *Drosophila* gene *peanut* (*pnut*) belongs to the Septin family of proteins and is required for cytokinesis (Neufeld & Rubin, 1994). RNAi-mediated depletion of *pnut* in the wing disc causes cytokinesis failure that leads to apoptosis and growth defects. We have reported that the oncogene *yki* is able to induce tumorigenesis in

---

*Gal4* to mark the dorsal compartment. The RFP area was used to determine the size of the dorsal compartment for "*bantam-sponge*" and "*bantam-sponge+trbl-RNAi+hid-RNAi*." Sizes of the dorsal compartment are normalized to the mean of the control. Statistical significance was determined by unpaired *t* test (n = 15; ****$P < 0.0001$). Larvae were kept constantly at 25°C.

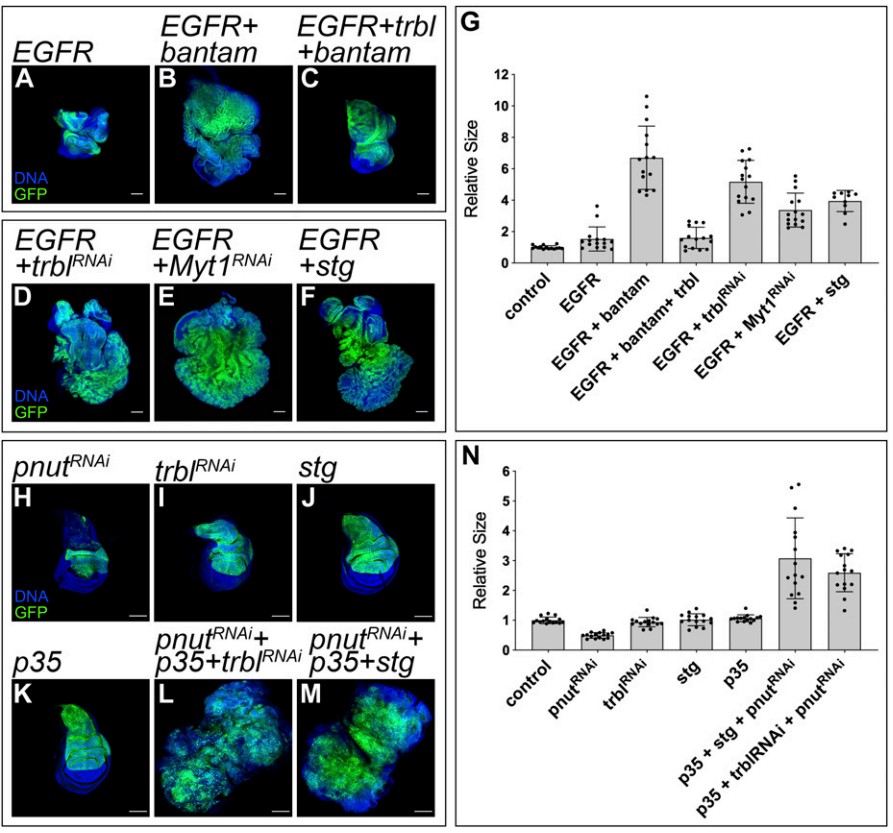

**Figure 5. Tumor suppression by *trbl*.**
**(A–F)** Confocal micrographs of third-instar wing imaginal discs. The indicated genes were expressed under the control of the *ap-Gal4* driver (dorsal compartment). The discs also expressed *UAS-GFP* to label the region where the corresponding transgenes were induced (green). DAPI was used to label the nuclei (blue). *ap-Gal4:UAS-GFP* was used as control (A). Scale bar: 100 μm. **(G)** Quantification of GFP-positive area in third-instar wing imaginal discs. GFP-positive area is normalized to the mean of the control. **(H–M)** Confocal micrographs of third-instar wing imaginal discs. The indicated genes were expressed under the control of the *ap-Gal4* driver (dorsal compartment). The discs also expressed *UAS-GFP* to label the region where the corresponding transgenes were induced (green). DAPI was used to label the nuclei (blue). Scale bar: 100 μm. **(N)** Quantification of GFP-positive area in third-instar wing imaginal discs. GFP-positive area is normalized to the mean of the control.

discs with reduced *pnut* (Gerlach et al, 2018). The oncogenic role of Yki in this context appears to be conserved and expression of YAP, the human ortholog of *yki*, also triggers tumorigenesis in human cells with cytokinesis failure (Ganem et al, 2014).

The fact that *bantam* is a central Yki target led us to analyze whether *trbl* regulation contributes to tumorigenesis in cells knocking down *pnut*. Wing discs expressing *UAS-pnut-RNAi* activate the c-Jun N-terminal kinase (JNK) pathway, which mediates their elimination by apoptosis. However, those cells accumulate in G2 and suppression of apoptosis is not sufficient to induce tumorigenesis (Gerlach et al, 2018). We tested whether *trbl* depletion could trigger tumorigenesis in cells depleting *pnut* and protected against apoptosis. Remarkably, simultaneous expression of *pnut-RNAi*, *p35*, and *trbl-RNAi* induced the formation of disorganized organs resembling tumors (Fig 5H–N). Comparable results were obtained when we expressed *cdc25-stg* as an alternative means of driving G2/M transition (Fig 5M and N). The expression of *p35* on its own did not affect tissue growth (Fig 5K and N).

In conclusion, we provide evidence showing that *trbl* down-regulation fuels tumorigenesis in two independent scenarios: EGFR-driven tumors, and discs depleting *pnut* and protected against apoptosis.

## Discussion

Here, we identify the cell cycle regulator *trbl* as a direct *bantam* target gene involved in the growth regulatory role of this miRNA.

Although *trbl* knock down accelerates G2/M progression, this does not produce obvious changes in tissue size. Two mechanisms might explain this apparent discrepancy. (1) Cells in the wing disc compensate for perturbations in the length of specific phases of the cell cycle by altering the duration of the other phases. This serves as a mechanism to keep normal division rates of cell proliferation, even when certain phases of the cycle are affected (Reis & Edgar, 2004). Faster G2/M progression, as a consequence of *trbl* depletion, would slow down G1/S to maintain a normal rate of cell division. (2) Perturbations in the cell cycle are normally accompanied by an increase in cell death (Karim & Rubin, 1998; Neufeld et al, 1998). Consistently, apoptosis can be detected in discs with reduced *trbl*, which might serve as a mechanism to prevent tissue overgrowth and tumor formation. Furthermore, knock down of the proapoptotic *bantam* target *hid*, combined with *trbl* depletion, induces tissue hyperplasia. Our results establish the minimal combination of *bantam* targets that, when depleted, drives tissue overgrowth. Tumors commonly combine increased rates of cell division with suppression of apoptosis (Green & Evan, 2002). We provide a new example, whereby defects in the cell cycle and apoptosis lead to tissue hyperplasia. In addition to *trbl* and *hid*, *bantam* represses other negative growth regulators (Brennecke et al, 2003; Herranz & Cohen, 2010; Herranz et al., 2012a, 2012b). Although individual depletion of those elements does not result in tissue hyperplasia, we cannot exclude them as important *bantam* targets in growth control that, together with *trbl* and *hid*, might contribute to the growth-promoting role of *bantam*.

The repressor of cytokine signaling, *Socs36E,* is one of the *bantam* target genes identified so far. *bantam* cooperates with *EGFR* in the formation of metastatic tumors, and *Socs36E* plays a central role in that process. *Socs36E* down-regulation causes an increase in JAK/STAT pathway activity that, in combination with EGFR, drives malignancy (Herranz et al, 2012b; Hombria & Serras, 2013). SOCS5 is the *Socs36E* human ortholog and, as observed in flies, SOCS5 acts as a tumor suppressor in EGFR/RAS–dependent cell transformation assays (Herranz et al, 2012b). Subsequent analyses found that the transforming activity of oncogenic Ras-V12 relies on its ability to down-regulate SOCS5/6 in human cell lines (Hong et al, 2014). The results presented here show that *trbl* limits the magnitude of growth of tumors co-expressing *EGFR* and *bantam*, and *trbl* down-regulation potentiates *EGFR*-driven tumor formation. This effect does not appear to be unique for *EGFR*-driven tumors and *trbl* is also required to limit tumorigenesis in cells with defective cytokinesis.

Cytokinesis failure can be tumorigenic, and it is proposed that around 40% of all human cancers have undergone a genome duplication event during their evolution (Zack et al, 2013). In *Drosophila*, cells with cytokinesis failure are eliminated by apoptosis as a consequence of JNK activation. Suppression of apoptosis is not sufficient to promote tumorigenesis as those cells accumulate in G2 and proliferate poorly (Gerlach et al, 2018). Additional signals are required to induce proliferation and tumorigenesis. Here, we show that *trbl* is one of those elements, and *trbl* depletion sufficed to trigger tumorigenesis in cells with cytokinesis failure protected against apoptosis.

*trbl* down-regulation is sufficient to drive tumorigenesis in wing discs expressing the oncogene *EGFR*. However, discs with cleavage defects also require suppression of apoptosis for tumor formation. Interestingly, the EGFR pathway promotes cell survival through *hid* down-regulation by Ras (Bergmann et al, 1998; Kurada & White, 1998). The role of EGFR in cell survival might explain why EGFR-driven tumors are not dependent on additional blockage of apoptosis, whereas discs down-regulating *pnut* and *trbl* require suppression of apoptosis for tumor formation. Studying whether Trbl plays similar roles in human tumor models might merit further investigation.

Previous studies have shown that different growth regulators such as Ras, dMyc, and dPI3K control tissue growth by regulating G1/S progression (Johnston et al, 1999; Weinkove et al, 1999; Prober & Edgar, 2000, 2002). Yki also controls G1/S progression by regulating the expression of genes, including *Cyclin E*, *E2F*, and *dMyc* (Wu et al, 2003; Goulev et al, 2008; Neto-Silva et al, 2010). In addition to that, Yki drives G2/M progression by regulating the transcription of the phosphatase *cdc25-stg* (Gerlach et al, 2018). miRNAs are typically embedded in regulatory loops to reduce noise and provide robustness to biological systems (Herranz & Cohen, 2010). Our results suggest the presence of a feed forward loop (FFL) downstream of Yki that includes the miRNA *bantam* and modulates G2/M transition. FFLs involve at least three elements: a regulator, X (Yki), which regulates Y (*bantam*), and gene Z (*cdc25-stg*), which is regulated by both X (Yki) and Y (*bantam*) (Fig 6). Both branches act in the same direction on Z (cdc25-stg), hence representing a coherent FFL (Alon, 2007). Yki drives tissue growth by inducing proliferation and repressing apoptosis. Interestingly, a

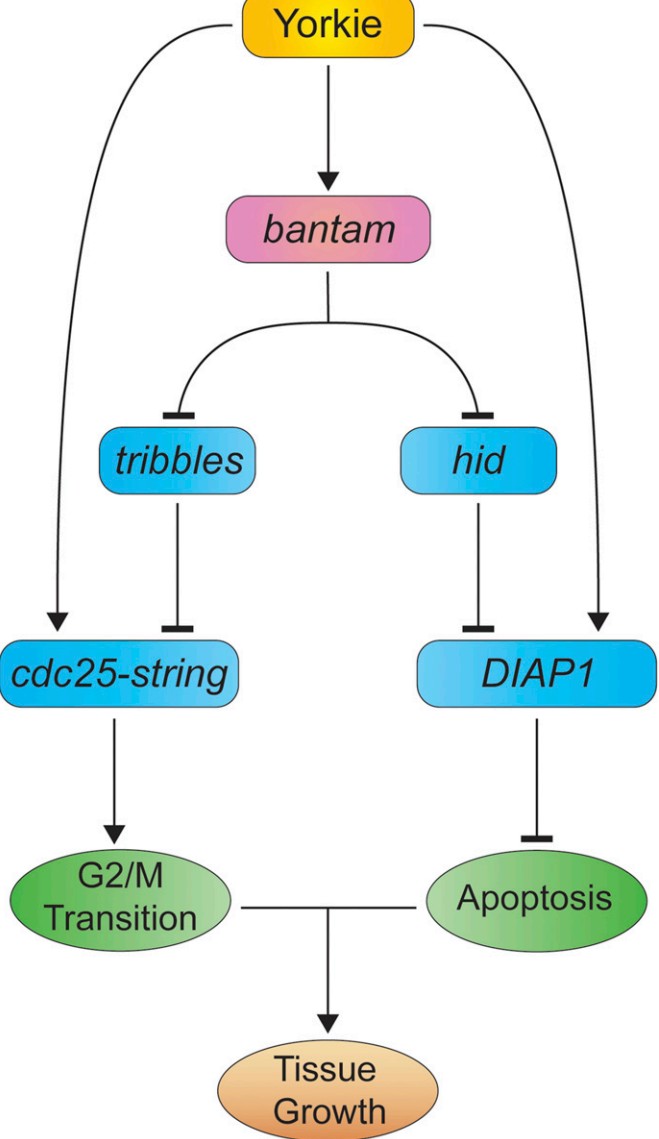

**Figure 6. Model illustrating the growth-promoting role of Yki and *bantam* in the wing epithelium.**
Yki promotes growth by inducing cell cycle progression and repressing apoptosis. Yki regulates G2/M progression directly by inducing the expression of its target gene *cdc25-stg*. Likewise, it limits apoptosis by inducing the expression of the repressor of apoptosis *DIAP1*. In parallel, Yki induces the expression of the miRNA *bantam*, which, in turn, represses *trbl* and *hid*. *trbl* acts as an inhibitor of *cdc25-stg*, and *hid* represses *DIAP1*. Yki triggers an FFL controlling cell cycle progression and apoptosis directly, and indirectly through *bantam*. The combination of these factors leads to tissue growth.

similar coherent FFL involving *bantam* is present in the "anti-apoptotic branch." Yki represses apoptosis directly by inducing the expression of *DIAP1*, and indirectly by inducing the expression of *bantam* that in turn represses the proapoptotic gene *hid* (Fig 6). These types of circuits can be used to reduce noise and ensure developmental decisions leading to an "all-or-none" outcome (Herranz & Cohen, 2010). This type of loop can serve to drive tumorigenesis, whereby sustained Yki activity would lead to the

activation of both arms in the FFL, resulting in an excess of proliferation and reduced apoptosis and consequently tumor formation. In fact, these kinds of loops have also been reported operating in human cancer (Iliopoulos et al, 2009).

# Materials and Methods

## *Drosophila* strains

The following *Drosophila* strains are described in the cited references: ap-Gal4 and nub-Gal4 (Calleja et al, 1996); ptc-Gal4 (Wilder & Perrimon, 1995); MS1096-Gal4 (Capdevila & Guerrero, 1994); bantam-Δ1, UAS-bantam-A, and UAS-bantam-D (Brennecke et al, 2003); EP3622 (Hipfner et al, 2002); UAS-miR-RHG (Siegrist et al, 2010); UAS-fly-FUCCI (Zielke et al, 2014); UAS-yki (Huang et al, 2005); UAS-EGFR (Buff et al, 1998); UAS-bantam-sponge and bantam-lacZ (P{lacW} banL1170a) (Herranz et al, 2012a); FRT-wtsX1 (Xu et al, 1995). Other stocks are described in FlyBase.

The following stocks were provided by the Bloomington *Drosophila* Stock Center (BDSC): UAS-p35 (#6298); UAS-stg (#56562); UAS-pnut-RNAi (#11791); UAS-Myt1-RNAi (#62892); and trbl-GFP (#61654).

The following stocks were provided by the Vienna *Drosophila* RNAi Center (VDRC): UAS-trbl-RNAi (#22114); UAS-hid-RNAi (#8269); and UAS-trbl-RNAi-KK (#106774).

The complete list of genotypes can be found as Supplemental Data 1.

## Transgene expression in the wing imaginal disc

The *apterous* and *patched* enhancers were used to express *Gal4* in the wing imaginal disc. To prevent early lethality due to transgene expression, the *tubulin-Gal80TS* construct was used for gene manipulation experiments. It allows the control of the timing of UAS transgene expression in a temperature dependent manner (Zeidler et al, 2004). *Drosophila* crosses were kept for 2 days at 18°C to lay eggs and larvae were kept for five additional days at 18°C. After that, the larvae were transferred to 29°C to induce transgene expression. Only third-instar wandering larvae were dissected to ensure that the animals analyzed were in the same developmental stage. Figure legends indicate when the timing and temperature differed from this procedure.

## *trbl* 3′UTR sensor

The *trbl* 3′UTR was amplified from cDNA (DGRC clone RH69304) with the primers *wt-fwd* and *wt-rev* and cloned into the tub-EGFP pCaSpeR4 sensor vector as an XhoI–NotI fragment. We generated a mutation in the *bantam* 3′ site, via PCR using the *wt-fwd* and *3′-mt-rev* primers, exchanging TC with AG. To mutate the *bantam* 5′ site (located more upstream in the *trbl* 3′UTR), we performed two PCRs with (1) the *wt-fwd* and *5′mt-rev* primers, exchanging AT with GC and (2) the *5′mt-fwd* and *wt-rev* primers, again exchanging AT with GC. We generated the full *trbl* 3′UTR with the *bantam* 5′ site mutated in a subsequent PCR using these overlapping fragments as templates

together with the *wt-fwd* and *wt-rev* primers. Using the same strategy but combining the *wt-fwd-* with the *3′-mt-rev* primers, we generated the *trbl* 3′UTR with both *bantam* sites mutated.

The following primers were used:

*wt-fwd*: 5′-GACTgcggccgcTTGGAGCTCGTGGAGTCACCC
*wt-rev*: 5′-TCAGctcgagCTTGGTGAGATCAAATTCCAATGTATTATGTTC TTT
*3′-mt-rev*: 5′-TCAGctcgagCTTGGTGActTCAAATTCCAATGTATTATGT TCTTTTTTGCAATTTTCACTT
*5′mt-rev*: 5'-CGACTGTCCTTTGATAAGgcCAACAGTTTGCAAAAAAATTA AAATTAAAATGTTAAATGT
*5′mt-fwd*: 5'-TGCAAACTGTTGgcCTTATCAAAGGACAGTCGCCC.

## Immunohistochemistry

The following primary antibodies were used: anti-ß-Gal (Developmental Studies Hybridoma Bank; 40-1a; mouse; 1:20); anti-Cas3-act (no. 9661; rabbit; 1:100; Cell Signaling Technology); anti-PH3 (no. 9701; rabbit; 1:100; Cell Signaling Technology). The following secondary antibodies were used: antimouse Alexa Fluor 555 (A-21425; goat; 1:400; Invitrogen); antimouse Alexa Fluor 647 (A-21237; goat; 1:400; Invitrogen); anti-rabbit Alexa Fluor 555 (A-21430; goat; 1: 400; Invitrogen); and antirabbit Alexa Fluor 647 (A-21246; goat; 1:400; Invitrogen).

Third-instar larvae were dissected in PBS. Fixation of the sample was performed with 4% formaldehyde solution for 20 min at room temperature. After that, the samples were washed three times for 10 min in phosphate-buffered saline-tween (PBT) and blocked for 20 min in PBT-BSA (BBT). The first antibody diluted in BBT was incubated with the samples at room temperature overnight followed by three 15-min washes in BBT. The secondary antibody together with DAPI in BBT was incubated with the sample for 2 h at room temperature followed by four 15-min washes in PBT. The samples were mounted in 90% glycerol with PBS containing 0.05% propyl gallate. Images of wing imaginal discs were acquired with a Leica SP8 confocal laser-scanning microscope and processed with ImageJ and Adobe Photoshop CC.

## Genetic mosaics

The FLP/FRT technique (Xu & Rubin, 1993) was used to generate *bantam* and *wts* loss of function clones. Second-instar larvae were heat-shocked for 1 h at 37°C. Next, the larvae were kept at 25°C. Clones were visualized by either loss of RFP or β-Gal expression.

## Flow cytometry

Dissection of wing imaginal discs of third-instar larvae was performed in PBS and trypsin–EDTA solution was used to dissociate the sample for 45 min. Cells were fixed with 4% formaldehyde solution for 20 min and stored in 70% ethanol for 2 h. The samples were incubated with DAPI in PBT overnight at 4°C. Flow cytometry was used to measure fluorescence of DAPI, GFP, and RFP in cells with a BD FACSAria Fusion. Doublet cells were excluded with an integral/ peak dot plot of DAPI fluorescence.

### Adult wing size and cell number quantification

The *MS1096* and *nubbin* enhancers were used to express *Gal*4 in the wing. Adult flies were fixed in glycerol-ethanol (25% glycerol, 75% ethanol) for three days and the wings were mounted in glycerol for imaging. Wing images of adult flies were taken with a Leica DM 5500B bright-field microscope. Tissue area measurement was performed with ImageJ. Bristle counting to determine cell density was obtained from high-magnification pictures of the region immediately anterior to the cross vein between veins four and five. The number of bristles in a 480 × 170 pixels area was quantified, and unpaired *t* test was used to determine statistical significance. Graphs were generated with GraphPad Prism 7.

### Wing imaginal disc size quantification

Eggs were collected on apple agar plates with yeast paste at 25°C for 24 h and larvae were kept on the plates for another 24 h. A maximum of 30 larvae were transferred to each vial with standard *Drosophila* food and stored for 4 days on 18°C. Then, the vials were switched to 29°C to activate transgene expression. Larvae were dissected as third-instar wandering larvae to ensure same developmental timing. Images were acquired with a Leica SP8 confocal laser-scanning microscope. The option "Threshold" was used to mark the whole GFP-positive area in ImageJ. After this, the option "Analyze Particles" was used to determine the area size. Unpaired *t* test was used to determine statistical significance and graphs were generated with GraphPad Prism 7. Figure legends indicate when the timing, temperature, or the fluorophore used for the area quantification differed from this procedure.

### Quantification of cells in mitosis

Third-instar wing imaginal discs were stained against PH3, which specifically labels mitotic cells. Images were acquired of the apical part of the wing disc with a Leica SP8 confocal laser-scanning microscope. An area of 250 × 250 pixels was used to count PH3-positive nuclei. Unpaired *t* test was used to determine statistical significance, and graphs were generated with GraphPad Prism 7.

### Quantification of apoptosis

Third-instar wing imaginal discs were stained against Cas3-act, which marks cells undergoing apoptosis. Images were acquired of the basal part of the wing disc with a Leica SP8 confocal laser-scanning microscope. An area of 250 × 250 pixels in the wing pouch was used to quantify the signal. The overall intensity of Cas3-act per genotype was quantified by using the measurement "Mean Grey Value" in ImageJ. Unpaired *t* test was used to determine statistical significance, and graphs were generated with GraphPad Prism 7.

### Western blot

15 third-instar larvae were transferred into protein extraction and immunoprecipitation buffer (RIPA; Sigma-Aldrich), cOmplete Protease Inhibitor Cocktail (Roche) added, and the larvae homogenized. The homogenate was centrifuged (15 min, 13,000 rpm, Eppendorf Centrifuge 5424, Rotor FA-45-24-11), and the supernatant used further. The BCA protein assay (Thermo Fisher Scientific) was used to determine the protein concentration. 10 µg of protein was loaded per sample for the Western blots. Antibody against GFP (ab13970, chicken; Abcam) was used to detect Trbl–GFP fusion protein. Antibody against Actin (A5441, mouse; Sigma-Aldrich) was used as a control. Secondary HRP goat antichicken (A16054; Invitrogen) and goat antimouse (P0447; Dako) antibodies were used together with the Pierce ECL Western Blotting Substrate (Thermo Fisher Scientific) to visualize protein levels.

### EdU incorporation

Third-instar larvae were dissected in PBS and incubated in 300 mM EdU in PBS for 30 min. Fixation of the sample was performed with 4% formaldehyde solution for 20 min at room temperature. Following, the samples were washed three times for 10 min in PBT and blocked for 20 min in BBT. The Click-iT EdU Alexa Fluor 488 Imaging Kit (Invitrogen) was used, and the samples were incubated for 1 h with the reaction mix provided by the Imaging Kit followed by one BBT wash for 10 min. The samples were incubated with DAPI in BBT for 2 h at room temperature followed by four 15-min washes with PBT. The samples were mounted in 90% glycerol with PBS containing 0.05% propyl gallate.

## Acknowledgements

We thank Carlos Estella for reagents; Melissa Alexandra Visser for technical support; Jan Michael Kugler for technical guidance; Olive Benét Kersey for comments on the manuscript; and the DSHB, VDRC, and BDSC for antibodies and fly strains. This work was supported by Novo Nordisk Foundation (grant number NNF0052223), a grant by the Neye Foundation for genetic models for cancer gene discovery, and a grant by Læge Sofus Carl Emil Friis og Hustru Olga Doris Friis' Legat.

### Author Contributions

SU Gerlach: conceptualization, data curation, formal analysis, validation, investigation, visualization, methodology, and writing—original draft, review, and editing.
M Sander: data curation, formal analysis, investigation, methodology, and writing—review and editing.
S Song: conceptualization, formal analysis, investigation, visualization, and methodology.
H Herranz: conceptualization, resources, supervision, funding acquisition, investigation, methodology, project administration, and writing—original draft, review, and editing.

### Conflict of Interest Statement

The authors declare that they have no conflict of interest.

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
