## [Reviewer comments · Life Science Alliance]

Life Science Alliance

The miRNA bantam regulates growth and tumorigenesis by repressing the cell cycle regulator tribbles

Stephan Gerlach, Moritz Sander, Shilin Song, and Héctor Herranz

DOI: <https://doi.org/10.26508/lsa.201900381>

Corresponding author(s): Héctor Herranz, University of Copenhagen

Review Timeline:

Submission Date:	2019-03-13
Editorial Decision:	2019-04-16
Revision Received:	2019-06-26
Editorial Decision:	2019-07-15
Revision Received:	2019-07-15
Accepted:	2019-07-15

Scientific Editor: Andrea Leibfried

Transaction Report:

April 16, 2019

Re: Life Science Alliance manuscript #LSA-2019-00381

Dr. Héctor Herranz
University of Copenhagen
Institute of Cellular and Molecular Medicine
Panum Institute.
Blegdamsvej, 3a - room 18.4.50
Copenhagen 2200
Denmark

Dear Dr. Herranz,

Thank you for submitting your manuscript entitled "The miRNA bantam regulates growth and tumorigenesis by repressing the cell cycle regulator tribbles" to Life Science Alliance. The manuscript was assessed by expert reviewers, whose comments are appended to this letter.

As you will see, the reviewers appreciate your data and support publication of a revised version of your manuscript here. The concerns raised by the reviewers all seem straightforward to address, and we would thus like to invite you to submit such a revised version to us. We would be happy to discuss the individual revision points further with you should this be helpful.

Thank you for this interesting contribution to Life Science Alliance. We are looking forward to receiving your revised manuscript.

Sincerely,

Andrea Leibfried, PhD

Executive Editor
Life Science Alliance
Meyerhofstr. 1
69117 Heidelberg, Germany
t +49 6221 8891 502
e a.leibfried@life-science-alliance.org
www.life-science-alliance.org

B. MANUSCRIPT ORGANIZATION AND FORMATTING:

Reviewer #1 (Comments to the Authors (Required)):

In their paper Gerlach and colleagues determine that the cell cycle regulator tribbles is a new target of the miRNA bantam, which in *Drosophila* has extensively been shown to control growth. The author use the wing imaginal disc system to show that a sensor that includes predicted bantam sites present in the tribbles 3'UTR is controlled by bantam and that tribbles modulation affects proliferation downstream of bantam and the hippo pathway component yorkie (YAP in humans).

The finally present evidence tribbles is part of a circuit regulate by yorkie and bantam that controls tissue growth by coordinating cell cycling and death decisions.

The genetic control of growth is central to tissue development, homeostasis and their alterations, which are observed in cancer and congenital syndromes. The fly system has played a key role in discovering most of the conserved growth regulators, such as components of the Tor and Hippo pathway among others. Since its serendipitous discovery, which sparked the entire miRNA field, bantam regulation of growth has represented a paradigm of posttranscriptional regulation. The work presented in the manuscript builds on previous discovery of the same authors and of others and represent an important step in revealing mechanisms that control coordination of cell cycle and cell death regulation. The manuscript presents an elegant set of genetic experiments which are solid, well controlled and properly quantified. It is clearly written and figure displays are logically organized. I only have few minor comments.

1-Some of the effect presented have been observed before in the same or in other systems (tumor suppressor activity of yorkie, cell cycle regulation by tribbles). These would probably make more sense as supplementary figures.

2-Bantam was reported to regulate many targets. However the authors have shown that 2 of them (tribbles and hid) are sufficient to recap most of the phenotype. Would you speculate in the discussion about the function of other targets? Is it tissue specificity? Is it other functions behind growth?

3- The first part of the discussion essentially restate most of what has been presented in the intro. Authors show streamline that part.

Reviewer #2 (Comments to the Authors (Required)):

In this manuscript, Gerlach et al use the *Drosophila* wing epithelium to identify genes required for tissue growth downstream of the Yki target gene bantam. The authors identify the G2/M regulator Tribbles as a target of the bantam miRNA. Although tribbles downregulation did not result in overgrowth of the wing imaginal primordium, the authors nicely show that together with hid downregulation, tribbles depletion is capable of driving overgrowth to a similar extent as bantam does. The authors suggest that Yki promotes proliferation and apoptosis inhibition both directly by regulating cell cycle regulators and Diap1 as well as indirectly through bantam, which represses both tribbles and hid. Additionally, the authors demonstrate that tribbles can act as a tumour suppressor in other tumorigenic conditions such as cytokinesis blockage.

Overall, the work is of a high standard and provides interesting insights into bantam-dependent growth regulation during normal *Drosophila* wing disc development as well as in some tumorigenic conditions. The authors convincingly demonstrate regulation of tribbles by bantam and show that together with hid is the minimal combination that can explain most of the overgrowth that bantam overexpression drives. However, there are several important issues as detailed below that the authors should address prior to publication of this manuscript.

Major points:

1. Figure 1 requires quantification of disc size since the bantam+CF and Yki+CF conditions seem to have large variability (compare Figure 1 with Figure 2 examples). Besides, the authors specify in the

methods section that "larvae were kept for 3 to 5 days at 29{degree sign}C" which can cause large differences in tissue size. Quantification of several examples would be helpful to support authors statement that "The effect of bantam in discs with cytokinesis failure was similar to the one observed by Yki expression (Fig 1C, I; and Fig 2G, H)." Figure 7H-M also needs quantification.

2. In Figure 3H-I the authors show bantam mutant clones in a Minute/+ background to demonstrate that endogenous bantam regulates the trbl 3'UTR sensor. Since whether endogenous bantam levels regulate tribbles is an important point in the manuscript, I believe that is necessary to show that Minute heterozygous cells do not show changes in the trbl-sensor (wild type clones in a minute/+ background).

3. In Figure 3D-I it would be important to show whether the regulation of the trbl 3'UTR sensor results in increased levels of Trbl protein (either with antibody staining or the trbl-GFP line).

4. To link trbl with Hippo pathway perturbation, the authors should check if the trbl sensor and/or trbl-GFP are reduced in wts or hpo mutant clones.

5. Does combined loss of trbl and hid rescue the bantam clonal undergrowth phenotype? If so, it would strengthen the idea that these are key downstream targets of ban.

6. In Figure 4 what is the effect of trbl depletion on cell number/tissue size?

7. In Figure 4, the FACS data show that trbl overexpression causes cells to accumulate in G2 (increase number of cells in G2 and decrease number of cells in G1). Does this result in fewer mitotic cells? Please show PH3 staining in this condition. I also wonder how the authors explain that this manipulation does not affect tissue size, especially since it was shown that trbl overexpression in the eye and wing imaginal discs shows defects (see Figure 5D in Price et al., 2002 and Figure 1 in Das et al., 2014). They mention Mata et al., 2000 to support the fact that trbl overexpression does not cause size defects, however that observation still lacks proper quantification. In fact, Mata et al., 2000 shows that trbl overexpression does affect cell size and cell number in the wing disc. I believe that authors should clarify this issue since it is important to validate their rescue experiment in the EGFR+bantam tumorigenic condition.

8. Figure 6F-G requires quantification. Although the trbl-RNAi+hid-RNAi wing size result is quite convincing, the number of apoptotic cells seems to be very low to explain the fact that trbl knock down did not result in tissue overgrowth. The same holds true for the example shown in FigS3A, in which many cas3-act positive cells seem to be in the ventral compartment. The increase in cell death associated with changes in cell proliferation cited in Neufeld et al., 1998 is much stronger than the one authors observe with Trbl downregulation. The authors should further test whether hid-RNAi rescues the apoptosis induced by trbl-RNAi.

9. In Figure 6H-L, authors use UAS-p35 to show that suppression of apoptosis in trbl-RNAi led to tissue overgrowth. I would strongly encourage the authors to carry out the same experiment using another way to block apoptosis (such as UAS-diap1). Overexpression of p35 leads to the presence of "undead cells" that are known to secrete several growth-promoting factors such as Wg and this might be the explanation for the fact that the p35+trbl-RNAi gives much higher relative size than the hid-RNAi+trbl-RNAi. Using UAS-diap1 would be more similar to the situation of hid-RNAi (no undead cells).

10. The authors suggest that apoptosis in response to trbl depletion can be used as a homeostatic

mechanism to offset tissue overgrowth. Is this specific to *trbl* or can it be mimicked by *dMyt1-RNAi* or *Stg*?

Minor points:

1. Authors should discuss the possible reasons for the difference between EGFR-activation and CF in the ability of *trbl* depletion to drive tumorigenesis (in CF, UAS-*p35* is required).
2. The authors consistently (several times in the text as well as in Figure 7 title) claim that *trbl* acts as a tumour suppressor in the wing imaginal disc. However, this conclusion might be interpreted as if depletion of *trbl* on its own results in tumour formation, which clearly is not the case. I feel that authors should be careful in specifying that *trbl* acts as a tumour suppressor in the wing imaginal disc only in EGFR activation condition. Moreover, the case with CF is quite different, because *trbl* depletion alone is not capable of driving tumorigenesis in a CF background unless *p35* is expressed as well.
3. From the observations in Figures 4 and 5 the authors say that "... these results suggest that *bantam* acts through *trbl* to stimulate tissue growth". I would be slightly more conservative and say "partially through *trbl*" since the rescue does not seem to be complete (show statistics between *bantam+trbl* and control in Figure 5G).

Methods section:

- Please specify the product number of each antibody listed as well as the concentrations at which each one was used.
- Please specify which secondary antibodies were used as well as the respective concentrations.
- Describe how the flow cytometry-based cell cycle analysis was performed.

Typos:

- In the summary blurb: "...which is involved in the *bantam*..."

Reviewer #3 (Comments to the Authors (Required)):

The submitted article demonstrates that *bantam* regulates growth by repressing *tribbles*. Previous work from the same lab has shown that coexpression of the *Yki* targets *string* and *DIAP1* can promote tumorigenesis in wing discs with defective cytokinesis. Now, they identify the *string* regulator *tribbles* as a direct *bantam* target gene and show that *tribbles* regulation by *bantam* is central in controlling tissue growth and tumorigenesis. The study is well designed and performed, and provides valuable insight on the very complex mechanistic bases of *bantam*'s role in development and disease. I recommend publication.

Considering/addressing the following points would result in a much more solid and rigorous article.

- 1- I really do not see the need for the CF model in this work. A great deal of the value of the main finding reported in the submitted manuscript is that it may be expected to apply to all processes that require cells to successfully cycle. That includes normal development as much as tumour growth, which itself includes CF-related tumours as much as any other type of tumour. Thus, quite frankly, the paper would be just as important -and much more clear- if it was limited to Figs 5, 6, S1 and S2, which are the ones that make the point. Reference to CF tumours do not contribute

anything to this story.

As a matter of fact, the other figures are fairly dispensable. Demonstrating that tissue overgrowth can be limited by compromising cell cycle progression is rather trivial as trivial is showing the opposite. More so when what goes for Yki, CF, etc is likely to go for any other tumour condition.

This is important because headings like "Trbl represses tumorigenesis in cells with cytokinesis failure" can be misleading if the point is not made clear that, more than likely, Trbl represses any kind of tumorigenesis.

2- Somewhat related to the above, the entire section "trbl antagonizes G2/M transition in the wing disc " is unnecessary because it quite simply confirms the obvious. Could be supplemental if anything.

3- The Becam et al., 2011 manuscript is cited, but not properly discussed. There, the authors showed that expression of a dsRNA form of tribbles did not rescue the defects in DV boundary formation caused by ectopic expression of bantam and concluded that bantam targets other molecular effectors involved in the maintenance of the DV affinity boundary. The submitted results open a question mark on these results: what effect could be expected by expressing dsRNA tribbles in cells that overexpress bantam (tribbles would be downregulated any how!) I was surprised to find out that this point is not discussed at all -despite numerous references to the Becam paper-.

4- "Aneuploidy is common in human cancers, reviewed in (Gordon et al., 2012) ". Aneuploidy is indeed very common in some human cancers, but it also not common at all in others that are just as malignant. There is ample published evidence in this regard that is not referred to, all too often. If the authors wish to discuss aneuploidy and cancer, they should provide the reader with a more comprehensive view of the subject.

5- I am confused by the reference to Dekanty et al., 2012: "In Drosophila, it [aneuploidy] triggers tumorigenesis (Dekanty et al., 2012)".

If I am not mistaken what Dekanty and colleagues claimed was that "chromosomal instability leads to an apoptotic response", i.e. in Drosophila, aneuploidy triggers cell death, which is not quite the same. Please, correct this mistake. Please, cite bibliography showing that aneuploidy does not trigger tumorigenesis in wing discs (Poulton et al. 2014 10.1016/j.devcel.2014.08.007, and others). Also, please, cite the very relevant recent paper from the Oliveira lab (Mirkovic 2019: <https://doi.org/10.1371/journal.pbio.3000016>) showing that other Drosophila cells respond differently to wing disc cells.

6-"the signals driving tumorigenesis in tetraploid cells".

Do the authors know for certain that tumourigenesis starts in tetraploid cells? In the Gerlach et al., 2018 article they found that "tumours induced by yki in a context of flawed cytokinesis were heterogeneous and composed of aneuploid cells of different sizes (Gerlach et al., 2018)" (i.e. not just tetraploids).

Indeed, cytokinesis failure causes tetraploid cells to begin with, and all sorts of polyploidy and aneuploid karyotypes later on. In the absence of solid evidence to substantiate that tumours start from tetraploid cells only, I would suggest avoiding statements that take that as a proven fact.

Reviewer #1 (Comments to the Authors (Required)):

In their paper Gerlach and colleagues determine that the cell cycle regulator tribbles is a new target of the miRNA bantam, which in *Drosophila* has extensively been shown to control growth. The author use the wing imaginal disc system to show that a sensor that includes predicted bantam sites present in the tribbles 3'UTR is controlled by bantam and that tribbles modulation affects proliferation downstream of bantam and the hippo pathway component yorkie (YAP in humans). The finally present evidence tribbles is part of a circuit regulate by yorkie and bantam that controls tissue growth by coordinating cell cycling and death decisions.

The genetic control of growth is central to tissue development, homeostasis and their alterations, which are observed in cancer and congenital syndromes. The fly system has played a key role in discovering most of the conserved growth regulators, such as components of the Tor and Hippo pathway among others. Since it serendipitous discovery, which sparked the entire miRNA field, bantam regulation of growth has represented a paradigm of posttranscriptional regulation. The work presented in the manuscript builds on previous discovery of the same authors and of others and represent an important step in revealing mechanisms that control coordination of cell cycle and cell death regulation. The manuscript presents an elegant set of genetic experiments which are solid, well controlled and properly quantified. It is clearly written and figure displays are logically organized. I only have few minor comments.

1-Some of the effect presented have been observed before in the same or in other systems (tumor suppressor activity of yorkie, cell cycle regulation by tribbles). These would probably make more sense as supplementary figures.

We show the effect of yki overexpression in Fig 3E. Even though we agree with the reviewer that this has been observed before, presenting it allows a direct comparison with discs that coexpress yki and trbl. We therefore believe it is important showing an example, as well as the size quantification, of those genetic manipulations in a main figure.

Previous studies have shown that *trbl* overexpression (gain of function approach – GOF) targets *stg* to degradation and therefore limits G2/M progression in the wing disc (Mata et al., 2000 - PMID: 10850493). However, the function of endogenous *trbl* in the proliferating wing epithelium has not yet been established. Here, we find that *trbl* downregulation leads, indeed, to faster G2/M transition. Given that GOF experiments can lead to non-specific effects, loss of function (LOF) analyses are central to determine gene function in a specific organ. We believe that the analysis of the *trbl* LOF, and the results obtained from that, are novel and relevant, and therefore merits being presented in a main figure. Besides, it supports other observations reported in this manuscript indicating that *bantam* regulates cell cycle progression by controlling G2/M transition the wing disc by reducing *trbl* levels.

2-*Bantam* was reported to regulate many targets. However the authors have shown that 2 of them (*tribbles* and *hid*) are sufficient to recap most of the phenotype. Would you speculate in the discussion about the function of other targets? Is it tissue specificity? Is it other functions behind growth?

In the discussion part we mention that depletion of *trbl* and *hid* is the minimum combination leading to tissue overgrowth but we cannot rule out that other *bantam* target genes might partially contribute to the growth regulatory role of *bantam*.

In the discussion it can be found as:

“Our results establish the smallest combination of bantam targets that, when depleted, drive tissue overgrowth. We provide a new example whereby defects in the cell cycle and apoptosis, biological processes commonly dysregulated in cancer, lead to tissue hyperplasia. In addition to trbl and hid, bantam represses other negative growth regulators (Brennecke et al., 2003; Herranz and Cohen, 2010; Herranz et al., 2012a; Herranz et al., 2012b). Although individual depletion of those elements does not result in tissue hyperplasia, we cannot exclude them as important bantam targets in growth control that, together with trbl and hid, might contribute to the growth-promoting role of bantam.”

Given that we do not have experimental evidence about potential role related to tissue specificity and/or other function different from growth regulation, we prefer not to discuss about this – it would mere speculation without any solid evidence supporting it.

3- The first part of the discussion essentially restate most of what has been presented in the intro. Authors show streamline that part.

We thank the reviewer for the suggestion. The mentioned first part of the discussion is rewritten and streamlined now.

Reviewer #2 (Comments to the Authors (Required)):

In this manuscript, Gerlach et al use the *Drosophila* wing epithelium to identify genes required for tissue growth downstream of the Yki target gene bantam. The authors identify the G2/M regulator Tribbles as a target of the bantam miRNA. Although tribbles downregulation did not result in overgrowth of the wing imaginal primordium, the authors nicely show that together with hid downregulation, tribbles depletion is capable of driving overgrowth to a similar extent as bantam does. The authors suggest that Yki promotes proliferation and apoptosis inhibition both directly by regulating cell cycle regulators and Diap1 as well as indirectly through bantam, which represses both tribbles and hid. Additionally, the authors demonstrate that tribbles can act as a tumour suppressor in other tumorigenic conditions such as cytokinesis blockage.

Overall, the work is of a high standard and provides interesting insights into bantam-dependent growth regulation during normal *Drosophila* wing disc development as well as in some tumorigenic conditions. The authors convincingly demonstrate regulation of tribbles by bantam and show that together with hid is the minimal combination that can explain most of the overgrowth that bantam overexpression drives. However, there are several important issues as detailed below that the authors should address prior to publication of this manuscript.

Major points:

1. Figure 1 requires quantification of disc size since the bantam+CF and Yki+CF conditions seem to have large variability (compare Figure 1 with Figure 2 examples). Besides, the authors specify in the methods section that "larvae were kept for 3 to 5 days at 29{degree sign}C" which can cause large differences in tissue size. Quantification of several examples would be helpful to support authors statement that "The effect of bantam in discs with cytokinesis failure was similar to the one observed by Yki expression (Fig 1C, I; and Fig 2G, H)." Figure 7H-M also needs quantification.

As suggested by the referee #3, we have eliminated the analysis of bantam modulation in the model of cytokinesis failure and therefore old Fig 1 and Fig 2 are not present in the revised version of our manuscript.

We have added the size quantification in new Fig 5, that corresponds to previous Fig 7.

2. In Figure 3H-I the authors show bantam mutant clones in a Minute/+ background to demonstrate that endogenous bantam regulates the trbl 3'UTR sensor. Since whether endogenous bantam levels regulate tribbles is an important point in the manuscript, I believe that it is necessary to show that Minute heterozygous cells do not show changes in the trbl-sensor (wild type clones in a minute/+ background).

Minute heterozygous cells do not show changes in the trbl-sensor. An example illustrating that control is shown in Fig S2 in the revised version of our manuscript.

3. In Figure 3D-I it would be important to show whether the regulation of the trbl 3'UTR sensor results in increased levels of Trbl protein (either with antibody staining or the trbl-GFP line).

We have not been able to obtain an aliquot of anti-Trbl. Pernille Rørth developed antisera against Trbl for their "Mata et al, 2000" paper but they ran out of antibody a long time ago.

We have used the Trbl-GFP to analyze trbl levels in bantam mutants. Trbl-GFP is inserted in the left arm of the 3rd chromosome (cytological location 77, 3-47%). bantam is located in the tip of the same chromosome arm (3-0,5%). This did not allow us to make this analysis in bantam mutant clones because bantam mutant clones would have two copies of trbl-GFP as compared to the surrounding heterozygous tissue that would only have one.

We agree with the reviewer that this is an important issue to address. Therefore, we have measured Trbl-GFP levels in bantam mutant larvae by western blot, using an anti-GFP. Bantam mutant larvae (transheterozygous for the bantam alleles bantam-delta1 and bantam EP3622) showed a robust increase in the levels of Trbl-GFP as

compared to control larvae. This is shown in Fig S1.

4. To link *trbl* with Hippo pathway perturbation, the authors should check if the *trbl* sensor and/or *trbl*-GFP are reduced in *wts* or *hpo* mutant clones.

Trbl-sensor is downregulated in *wts* mutant clones. This is shown in Fig 1J, K.

In the text: *“The Hippo pathway regulates the expression of bantam, which is central mediating the Yki growth regulatory role (Nolo et al., 2006; Thompson and Cohen, 2006). Hippo activation results in Warts (Wts) phosphorylation, which acts with its cofactor Mats to phosphorylate and inactivate Yki. wts mutant tissue upregulates Yki and overgrows, (reviewed in (Pan, 2010). As observed in discs upregulating bantam, wts mutant cells downregulated the trbl-sensor (Fig 1J, K). trbl 3’UTR is thus sensitive to Hippo pathway activity.”*

5. Does combined loss of *trbl* and *hid* rescue the *bantam* clonal undergrowth phenotype? If so, it would strengthen the idea that these are key downstream targets of *ban*.

Bantam mutant cells are eliminated by cell competition. Cell competition compares cellular fitness between different cells and triggers the elimination of the less fit population of cells. The competitive characteristics of clones of cells are defined not only by their proliferative potential, but also by other cellular mechanisms such as cellular polarity, cell size, metabolic status, mechanical tension, etc. *Bantam* mutant cells deregulate multiple targets that might compromise the competitive properties of those cells.

To analyze the contribution of the *bantam* target genes *trbl* and *hid* to the growth regulatory role of *bantam*, we compared disc size of discs depleting *bantam* in the dorsal compartment with discs depleting *bantam*+*hid*+*trbl* (there is not cell competition between different compartments in the wing disc). We used a UAS-*bantam*-sponge to downregulate *bantam* specifically in the dorsal compartment. Expression of *bantam*-sponge leads to a reduction in tissue size. Depletion of *hid* and *trbl* in discs expressing *bantam*-sponge

rescued the growth defects induced by bantam depletion. This strengthens the idea that these are key bantam targets. These results are shown in Fig 4I, K.

In the test: "Next, we studied the consequences of downregulating trbl and hid in discs with reduced bantam. miRNA sponges controlled by UAS sequences allow spatially-controlled downregulation of miRNAs under Gal4 control (Loya et al., 2009). The expression of a UAS-bantam-sponge transgene causes a reduction in tissue size (Herranz et al., 2012a). Notably, discs expressing simultaneously UAS-bantam-sponge, UAS-trbl-RNAi, and UAS-hid-RNAi were increased in size (Fig 4I-K). This suggests that codepletion of trbl and hid is sufficient to drive tissue overgrowth, even when bantam levels are reduced."

6. In Figure 4 what is the effect of trbl depletion on cell number/tissue size?

We have performed that analysis, which is shown in Fig 2H-K and Fig S4.

This is described text as: "We used the adult wing to analyze how trbl affected cell number and tissue size. In the adult wing, each hair-like structure corresponds to a trichome originating from a single epithelial cell. Trichome density can hence be used as a proxy of cell size and number. trbl depletion did not affect cell or wing size. trbl overexpression caused a mild reduction in wing size. These wings showed an increase in cell size revealed by reduced trichome density, indicating that those wings had fewer cells (Fig 2H-K, Fig S4). The increase cell size observed upon trbl upregulation is characteristic of cells with delayed cell cycle progression (Neufeld et al., 1998) and is consistent with the PH3 staining and the FUCCI-FACS analysis."

7. In Figure 4, the FACS data show that trbl overexpression causes cells to accumulate in G2 (increase number of cells in G2 and decrease number of cells in G1). Does this result in fewer mitotic cells? Please show PH3 staining in this condition. I also wonder how the authors explain that this manipulation does not affect tissue size, especially since it was shown that trbl overexpression in the eye and wing imaginal discs shows defects (see Figure 5D in Price et al., 2002 and Figure 1 in Das et al., 2014). They mention Mata et al., 2000 to support the fact that trbl overexpression does not cause size defects, however that

observation still lacks proper quantification. In fact, Mata et al., 2000 shows that *trbl* overexpression does affect cell size and cell number in the wing disc. I believe that authors should clarify this issue since it is important to validate their rescue experiment in the EGFR+bantam tumorigenic condition.

PH3 staining is shown in Fig 2E-G.

Described in the text as: *“We used anti-Phospho-Histone H3 (PH3) as a specific marker of cells in mitosis to study whether trbl overexpression affected the rate of cell proliferation of the wing disc. We detected a reduction in the number of PH3-positive cells in discs expressing UAS-trbl when compared control discs (Fig 2E-G). These observations are consistent with previous reports (Mata et al., 2000; Reis and Edgar, 2004) and show that trbl overexpression limits cell proliferation by dampening G2/M progression.”*

Our results show that *trbl* manipulation in an otherwise normal wing does not have a major impact in wing size. 5D in Price et al., 2002 shows an adult wing over-expressing *Dwee-1*, not *trbl*. In Das et al., 2014, the authors detect a reduction in wing size when over-expressing *trbl* under the control of *en-Gal4*. Although the results obtained by our manuscript and Das et al., 2014 is consistent, the magnitude in size reduction observed in Das et al., 2014 is bigger than the one we observed in our experiments. In Das et al., 2014, the authors used a different Gal4 driver (*en-Gal4*), which is not only expressed in the wing but in the posterior compartment of all the fly segments. The effect they observed might be partially explained by a systemic or non-autonomous effect of expressing *trbl* in different parts of the fly. In our manuscript, we preferred to use specific wing Gal4 drivers as *MS1096-Gal4* and *nub-Gal4*. In both cases, we find consistent results. Again, the magnitude is different to the one observed in Das et al., 2014, but the trend is the same and we can not find any inconsistency between their and our results.

8. Figure 6F-G requires quantification. Although the *trbl*-RNAi+*hid*-RNAi wing size result is quite convincing, the number of apoptotic cells seems to be very low to explain the fact that *trbl* knock down did not result in tissue overgrowth. The same holds true for the example shown in FigS3A, in which many *cas3-act* positive cells seem to be in the ventral compartment. The increase in cell death associated with changes in cell proliferation cited

in Neufeld et al., 1998 is much stronger than the one authors observe with *Trbl* downregulation. The authors should further test whether *hid*-RNAi rescues the apoptosis induced by *trbl*-RNAi.

We have added the requested quantification (see new Fig 4F-H).

hid-RNAi did not completely rescue the levels of apoptosis observed in discs depleting UAS-*trbl*-RNAi. This is shown in Fig S5G. In the text can be found as “*hid* depletion did not completely rescue the induction of apoptosis observed in discs expressing UAS-*trbl*-RNAi (Fig S5), suggesting that other proapoptotic genes, in addition to *hid*, might contribute to the apoptotic response to *trbl* knock down.”

9. In Figure 6H-L, authors use UAS-p35 to show that suppression of apoptosis in *trbl*-RNAi led to tissue overgrowth. I would strongly encourage the authors to carry out the same experiment using another way to block apoptosis (such as UAS-*diap1*). Overexpression of p35 leads to the presence of "undead cells" that are known to secrete several growth-promoting factors such as *Wg* and this might be the explanation for the fact that the p35+*trbl*-RNAi gives much higher relative size than the *hid*-RNAi+*trbl*-RNAi. Using UAS-*diap1* would be more similar to the situation of *hid*-RNAi (no undead cells).

We have used a UAS transgene expressing a miRNAs that simultaneously inhibits the proapoptotic genes *rpr*, *hid*, and *grim* (UAS-miR-RHG), and has been proven efficient blocking cell death (see, for example, PMID: 20346676; PMID: 30080872). Expression of this miRNA together with *trbl*-RNAi results in the formation of overgrown discs, which is consistent with the results we have obtained expressing p35+*trbl*-RNAi or *hid*-RNAi+*trbl*-RNAi. We have therefore obtained comparable results in 3 independent conditions: 1) *trbl*-RNAi + UAS-p35; 2) *trbl*-RNAi + UAS-*hid*-RNAi; and 3) *trbl*-RNAi + UAS-miR-RHG.

This is now shown in Fig S5.

10. The authors suggest that apoptosis in response to *trbl* depletion can be used as a homeostatic mechanism to offset tissue overgrowth. Is this specific to *trbl* or can it be mimicked by *dMyt1*-RNAi or *Stg*?

The effect of *trbl* downregulation can be mimicked by *stg* overexpression. This result is shown in Fig S6, and can be found in the text as, “*Similar results were obtained when we overexpressed cdc25-stg as an alternative way of forcing G2/M transition (Fig S6).*”

Minor points:

1. Authors should discuss the possible reasons for the difference between EGFR-activation and CF in the ability of *trbl* depletion to drive tumorigenesis (in CF, UAS-p35 is required).

We agree with the point of the reviewer and added this part to the discussion. It can be found as: “*trbl* downregulation is sufficient to drive tumorigenesis in discs expressing the oncogene EGFR. However, discs with cleavage defects also require suppression of apoptosis for tumor formation. Interestingly, EGFR pathway promotes cell survival through *hid* downregulation by Ras (Bergmann et al., 1998; Kurada and White, 1998). The role of EGFR in cell survival might explain why EGFR-driven tumors are not dependent on additional blockage of apoptosis, while discs downregulating *pnut* and *trbl* require suppression of apoptosis for tumor formation.”

2. The authors consistently (several times in the text as well as in Figure 7 title) claim that *trbl* acts as a tumour suppressor in the wing imaginal disc. However, this conclusion might be interpreted as if depletion of *trbl* on its own results in tumour formation, which clearly is not the case. I feel that authors should be careful in specifying that *trbl* acts as a tumour suppressor in the wing imaginal disc only in EGFR activation condition. Moreover, the case with CF is quite different, because *trbl* depletion alone is not capable of driving tumorigenesis in a CF background unless p35 is expressed as well.

We have carefully addressed that. In the revised version of the manuscript we have been more precise stating the specific context in which we have observed *trbl* behaving as a tumor suppressor.

3. From the observations in Figures 4 and 5 the authors say that "... these results suggest that bantam acts through trbl to stimulate tissue growth". I would be slightly more conservative and say "partially through trbl" since the rescue does not seem to be complete (show statistics between bantam+trbl and control in Figure 5G).

As suggested by the reviewer, we have added that.

Methods section:

- Please specify the product number of each antibody listed as well as the concentrations at which each one was used.

- Please specify which secondary antibodies were used as well as the respective concentrations.

As suggested by the reviewer, we have specified the primary and secondary antibodies as well as the product number and the concentrations at which each one was used.

- Describe how the flow cytometry-based cell cycle analysis was performed.

A description is included in the revised version of the manuscript.

Typos:

- In the summary blurb: "...which is involved in the bantam..."

Fixed, thanks!

Reviewer #3 (Comments to the Authors (Required)):

The submitted article demonstrates that bantam regulates growth by repressing tribbles. Previous work from the same lab has shown that coexpression of the Yki targets string and DIAP1 can promote tumorigenesis in wing discs with defective cytokinesis. Now, they identify the string regulator tribbles as a direct bantam target gene and show that tribbles regulation by bantam is central in controlling tissue growth and tumorigenesis. The study is well designed and performed, and provides valuable insight on the very complex mechanistic bases of bantam's role in development and disease. I recommend publication.

Considering/addressing the following points would result in a much more solid and rigorous article.

1- I really do not see the need for the CF model in this work. A great deal of the value of the main finding reported in the submitted manuscript is that it may be expected to apply to all processes that require cells to successfully cycle. That includes normal development as much as tumour growth, which itself includes CF-related tumours as much as any other type of tumour. Thus, quite frankly, the paper would be just as important -and much more clear- if it was limited to Figs 5, 6, S1 and S2, which are the ones that make the point. Reference to CF tumours do not contribute anything to this story.

As a matter of fact, the other figures are fairly dispensable. Demonstrating that tissue overgrowth can be limited by compromising cell cycle progression is rather trivial as trivial is showing the opposite. More so when what goes for Yki, CF, etc is likely to go for any other tumour condition.

This is important because headings like "Trbl represses tumorigenesis in cells with cytokinesis failure" can be misleading if the point is not made clear that, more than likely, Trbl represses any kind of tumorigenesis.

Following the suggestion by the reviewer, we have eliminated Figs 1 and 2 from the revised version of our manuscript. In those figures, we showed the role of bantam in the CF model.

We have, however, maintained Fig 6 showing that *trbl* limits tumorigenesis in two different tumor models: 1) EGFR-driven tumors and 2) tumors depleting *pnut* and protected against apoptosis. We completely agree with the reviewer that *Trbl* might well repress any kind of tumorigenesis. However, the observation that compromising the cell cycle (by *trbl* modulation) does not have a major impact in normal growth but has a strong effect in the formation of tumors in the wing disc. This suggests that, while normal cells use mechanisms that compensate for defects in the cell cycle and maintain normal tissue size, perturbed cell cycle might have profound effects in the context of tumor formation. We therefore believe that reporting the different consequences of *trbl* manipulation (or cell cycle deregulation by other independent means), in normal and tumor growth is relevant in this context.

2- Somewhat related to the above, the entire section "*trbl* antagonizes G2/M transition in the wing disc " is unnecessary because it quite simply confirms the obvious. Could be supplemental if anything.

Previous studies have shown that *trbl* overexpression targets *stg* to degradation and therefore limits G2/M progression in the wing disc (Mata et al., 2000 - PMID: 10850493). However, the function of endogenous *trbl* in the proliferating wing epithelium has not yet been established. Here we find that *trbl* downregulation leads indeed to faster G2/M transition. This is consistent with previous observation and we agree with the reviewer in that the observation that endogenous *trbl* antagonizes G2/M transition might seem obvious. That said, we believe that this result is relevant. This has not been previously studied and supports other observations presented in this manuscript indicating that bantam regulates cell cycle progression, through *trbl*, by controlling G2/M transition in the wing disc.

3- The Becam et al., 2011 manuscript is cited, but not properly discussed. There, the

authors showed that expression of a dsRNA form of tribbles did not rescue the defects in DV boundary formation caused by ectopic expression of bantam and concluded that bantam targets other molecular effectors involved in the maintenance of the DV affinity boundary. The submitted results open a question mark on these results: what effect could be expected by expressing dsRNA tribbles in cells that overexpress bantam (tribbles would be downregulated any how!) I was surprised to find out that this point is not discussed at all -despite numerous references to the Becam paper-.

Becam and cols identified the gene *enabled* as a *bantam* target involved in the maintenance of the DV boundary of the wing imaginal disc. In our manuscript, we do not evaluate the impact of the *bantam-trbl* relationship in the formation of the DV boundary. We rather focus in the role of bantam in growth control and tumorigenesis, and not in the functions of the *bantam-trbl* interaction in the generation and/or maintenance of the DV boundary. We cited that paper as an example of *bantam* target genes identified in the past but we prefer not to discuss this paper because the regulation of boundary formation is out of the scope of our manuscript.

4- "Aneuploidy is common in human cancers, reviewed in (Gordon et al., 2012) ".

Aneuploidy is indeed very common in some human cancers, but it also not common at all in others that are just as malignant. There is ample published evidence in this regard that is not referred to, all too often. If the authors wish to discuss aneuploidy and cancer, they should provide the reader with a more comprehensive view of the subject.

We fully agree with the reviewer in that, although aneuploidy is commonly observed in human cancers, it remains still unclear whether it is a cause or consequence of cancer. Given that the main focus of this work is describing the role of the gene tribbles as relevant bantam target in growth control and tumorigenesis, we have removed that part.

5- I am confused by the reference to Dekanty et al., 2012: "In Drosophila, it [aneuploidy] triggers tumorigenesis (Dekanty et al., 2012)".

If I am not mistaken what Dekanty and colleagues claimed was that "chromosomal instability leads to an apoptotic response", i.e. in Drosophila, aneuploidy triggers cell

death, which is not quite the same. Please, correct this mistake. Please, cite bibliography showing that aneuploidy does not trigger tumorigenesis in wing discs (Poulton et al. 2014 10.1016/j.devcel.2014.08.007, and others). Also, please, cite the very relevant recent paper from the Oliveira lab (Mirkovic 2019: <https://doi.org/10.1371/journal.pbio.3000016>) showing that other Drosophila cells respond differently to wing disc cells.

As mentioned in the previous point, that part has been removed in the revised version of the manuscript.

6-"the signals driving tumorigenesis in tetraploid cells".

Do the authors know for certain that tumourigenesis starts in tetraploid cells? In the Gerlach et al., 2018 article they found that "tumours induced by yki in a context of flawed cytokinesis were heterogeneous and composed of aneuploid cells of different sizes (Gerlach et al., 2018)" (i.e. not just tetraploids).

Indeed, cytokinesis failure causes tetraploid cells to begin with, and all sorts of polyploidy and aneuploid karyotypes later on. In the absence of solid evidence to substantiate that tumours start from tetraploid cells only, I would suggest avoiding statements that take that as a proven fact.

We have removed that statement in the revised version of the manuscript.

July 15, 2019

RE: Life Science Alliance Manuscript #LSA-2019-00381R

Dr. Héctor Herranz
University of Copenhagen
Institute of Cellular and Molecular Medicine
Panum Institute.
Blegdamsvej, 3a - room 18.4.50
Copenhagen 2200
Denmark

Dear Dr. Herranz,

Thank you for submitting your revised manuscript entitled "The miRNA bantam regulates growth and tumorigenesis by repressing the cell cycle regulator tribbles". As you will see, the reviewers appreciate the introduced changes, and we would thus be happy to publish your paper in Life Science Alliance pending final revisions necessary to meet our formatting guidelines:

- please add a callout to Fig 5H,I,J in the manuscript text.
- the scale bars are not always easily visible (e.g. Fig 1D,I), please check your figures one more time to enhance visibility where needed.
- please move the supplementary legends and reference into the main ms file, the genotype list can be included as supplementary information S1 and should be referred to in the manuscript text (methods section).

A. FINAL FILES:

B. MANUSCRIPT ORGANIZATION AND FORMATTING:

Sincerely,

Reviewer #1 (Comments to the Authors (Required)):

the revised manuscript address all my points and those of other referees and thus I judge it suitable for publication in the current form.

Reviewer #2 (Comments to the Authors (Required)):

The authors have satisfactorily addressed my comments and I am happy to recommend publication in LSA.

July 15, 2019

RE: Life Science Alliance Manuscript #LSA-2019-00381RR

Dr. Héctor Herranz
University of Copenhagen
Institute of Cellular and Molecular Medicine
Panum Institute.
Blegdamsvej, 3a - room 18.4.50
Copenhagen 2200
Denmark

Dear Dr. Herranz,

Thank you for submitting your Research Article entitled "The miRNA bantam regulates growth and tumorigenesis by repressing the cell cycle regulator tribbles". It is a pleasure to let you know that your manuscript is now accepted for publication in Life Science Alliance. Congratulations on this interesting work.

DISTRIBUTION OF MATERIALS:

Again, congratulations on a very nice paper. I hope you found the review process to be constructive and are pleased with how the manuscript was handled editorially. We look forward to future exciting

submissions from your lab.

Sincerely,
